# Impact of the COVID-19 Pandemic on Clinical Supervision of Healthcare Students in Rural Settings: A Qualitative Study

**DOI:** 10.3390/ijerph19095555

**Published:** 2022-05-03

**Authors:** Priya Martin, Lucylynn Lizarondo, Geoff Argus, Saravana Kumar, Srinivas Kondalsamy-Chennakesavan

**Affiliations:** 1Rural Clinical School, Faculty of Medicine, The University of Queensland, Toowoomba, QLD 4350, Australia; s.kondalsamychennakes@uq.edu.au; 2Cunningham Centre, Darling Downs Health, Toowoomba, QLD 4350, Australia; 3JBI, The University of Adelaide, Adelaide, SA 5001, Australia; lucylynn.lizarondo@adelaide.edu.au; 4Southern Queensland Rural Health, The University of Queensland, Toowoomba, QLD 4350, Australia; g.argus@uq.edu.au; 5School of Psychology and Counselling, University of Southern Queensland, Toowoomba, QLD 4350, Australia; 6Allied Health and Human Performance, University of South Australia, Adelaide, SA 5001, Australia; saravana.kumar@unisa.edu.au

**Keywords:** student placements, clinical supervision, rural health, COVID-19 pandemic

## Abstract

The COVID-19 pandemic has caused significant disruptions to healthcare student placements worldwide, including already challenged rural areas in Australia. While accounts are emerging of student experiences in larger centers and from a student perspective, there is a need for in-depth exploration of student supervisor experiences in rural areas at the onset of the pandemic. This study aims to address this gap through 23 individual, semi-structured interviews with healthcare workers from ten health professions who were either direct student supervisors or in roles supporting student supervisors A reflexive thematic analysis approach was used to develop four themes, namely compounding stress, negative impacts on student learning, opportunity to flex and innovate, and targeted transitioning support strategies. The findings indicate that healthcare workers with student supervision responsibilities at the onset of the pandemic experienced high levels of stress and wellbeing concerns. This study sheds light on the importance of supporting student supervisors in rural areas, and the need for implementing targeted support strategies for new graduates whose placements were impacted by the pandemic. This is not only essential for supporting the rural healthcare workforce but is also imperative for addressing inequalities to healthcare access experienced in rural communities.

## 1. Introduction

Clinical placements of prequalification healthcare students are an important means of ensuring their readiness for work following graduation. Student placements in Australia, similar to other parts of the world, were impacted significantly because of the COVID-19 pandemic. Queensland, an Australian state where this study was conducted, experienced its first wave of COVID-19 from early March 2020 [1]. Several placements across professions were cancelled at the onset of the pandemic [2,3]. Those placements that were able to continue underwent several changes including reduced access to the usual breadth and variety of clinical presentations, cancellations of outpatient services and elective surgeries, students needing to use personal protective equipment (PPE), and a change from face-to-face to telehealth service delivery formats [2,4]. Occasionally, clinical supervision of students also changed from face-to-face to telesupervision arrangements (i.e., remote supervision undertaken using technology) [4].

Global accounts of adverse effects of the pandemic on student learning and supervision during placements, its potential impact on their transition to practice, and the negative effects on healthcare worker and student mental health and wellbeing, are emerging [2,3,4,5,6,7,8]. A recent Australian mixed-methods study, including interviews with 29 students undertaking rural placements, reported negative effects of the pandemic on student learning, reduced supervision, and concerns about graduation and career prospects. In this study, lower levels of clinical supervision were noted to be detrimental to student learning [3]. Another national Australian survey of 1505 rural students from nursing, midwifery, medicine, and allied health, found that students that were female, and those younger in age, were more prone to reporting concerns related to their mental health and study progression [2]. Recent scoping reviews on the impact of the pandemic on healthcare workers health and wellbeing have highlighted the high levels of anxiety, depression, distress, and insomnia experienced by this group [7,8]. It has been further noted that female healthcare workers and nurses are among those that were at an increased risk of experiencing these impacts [9]. There is a lack of studies investigating in-depth experiences of student clinical supervisors and the impact of the pandemic on their supervision provision

Placements outside bigger metropolitan centers (i.e., regional, rural, and remote areas, referred to as ‘rural’ hereafter) can present unique challenges to students, supervisors, health services, and universities alike. Physical, emotional, and social isolation; lack of housing; financial burden; and commuting distances are noted as key barriers in the recent literature [10,11]. Student placements are an essential workforce recruitment strategy in these resource-constrained areas [12,13]. Therefore, disruptions to placements are likely to have a potential flow-on effect for graduate recruitment in these areas, which can be detrimental to healthcare access in these communities [14,15]. Given the disruptions caused by the COVID-19 pandemic, it is necessary to explore its effects on student clinical placements in these areas, in order to inform transition to support strategies for affected students, as well as to develop strategies to deal with similar unexpected events in the future. It is also essential to explore the impact this has had on healthcare workers who act as student supervisors, to understand workforce implications. This can inform the development of targeted strategies to better support this highly valued cohort. To address these gaps, this study explored healthcare workers experiences of the impact of the COVID-19 pandemic on clinical supervision provision during student placements.

## 2. Materials and Methods

### 2.1. Research Design

This study was designed using a qualitative constructivist lens, where meaning is co-created by participants and the interviewer [16].

### 2.2. Setting

This study was conducted in four rural health services in the public health sector in Queensland, Australia. Collectively, they service rural communities across their jurisdictions. The health services tend to use a hub and spoke model of service delivery with several services being coordinated from the larger centers, and consisting of drive-in and drive-out models to meet the healthcare needs of those living in smaller, remote towns. Senior healthcare professionals in all disciplines are expected to contribute to the education of the future healthcare workforce, primarily through provision of clinical education or student placement opportunities. Depending on the course undertaken and the university involved, the student placement length can vary from days to months. Disciplines have their own placement organization processes as well as central support staff to coordinate and support students on placements and their supervisors.

### 2.3. Participants and Sampling

Post-qualification healthcare professionals from allied health (including audiology, clinical measurements, exercise physiology, medical radiation, music therapy, nutrition and dietetics, occupational therapy, pharmacy, physiotherapy, podiatry, prosthetics and orthotics, psychology, social work, and speech pathology), nursing, midwifery, and medicine were invited to participate in the study. To be eligible, participants had to either be a student supervisor who supervised at least one student since the onset of the COVID-19 pandemic, or in a leadership, management, or support role with a responsibility for student supervision (also referred to as clinical education). Participants were recruited through a combination of sampling methods [17]:Combined or stratified sampling strategy, whereby participants from a preceding survey study were informed about this interview study and encouraged to contact the first author to express interest in being interviewed, or to obtain further information about the study.Snowball sampling strategy, whereby interviewees were asked for suggestions of other potential interviewees that would meet the inclusion criteria.Maximum variation sampling strategy. whereby interviewees were recruited across a broad spectrum of professional backgrounds and roles including new student supervisors, experienced student supervisors, and those holding student supervision portfolios in support, management, and leadership roles [17].

### 2.4. Procedure

Consenting participants were sent a copy of the interview guide (Appendix A) via email at least a day prior to the interview to encourage reflection. All interviews were conducted by the first author who is a qualitatively trained researcher, with extensive experience in student supervision and the conduct of interview studies. Given the dispersed locations, participants were able to choose between a face-to-face or Teams interview.

### 2.5. Data Collection

Semi-structured, individual interviews were used to collect data. The interview guide, informed by results from a preceding survey study, were piloted with two participants with expertise in student supervision. It contained a set of open-ended questions with associated prompts. The guide included varied questions depending on the work role of the interviewee (i.e., student supervisor or other roles). All interviews were audio-recorded with consent and transcribed by an independent typist. Interviews were conducted between May and July 2021.

### 2.6. Analysis

A reflexive thematic analysis method [18] was used to analyze the data. In reflexive thematic analysis, the researchers play an integral role in knowledge production, familiarizing themselves with the data, and continually questioning and querying assumptions while interpreting and coding the data. They exercise reflective and thoughtful engagement with the data and the analytic process [18]. Accordingly, two qualitative researchers (PM and LL, who are healthcare workers with a background in occupational therapy (PM) and physiotherapy (LL), content experts in clinical supervision, with extensive experience in student supervision, and qualitative research methods) analyzed the data independently and collaborated through regular discussions to construct themes.

### 2.7. Rigor

The researchers employed a variety of measures to enhance the trustworthiness of the study, including the use of reflexivity, using an iterative approach, ensuring information richness, piloting of the interview guide, and co-analysis and triangulation of the data by two researchers. Given the qualitative nature of this research, it is important to acknowledge the background of the first author (PM) as they may have, knowingly or unknowingly, influenced the data collection, analysis, and data interpretation processes. The first author (PM) is a locally renowned researcher with more than a decade of experience in the field of clinical supervision and it is likely they were known to some participants and had pre-existing working relationships. To mitigate the extent of these potential influences on participants’ perspectives and responses, the interviews followed a pre-established interview guide: The participants were asked to speak freely with confidentiality assured, and the first author had frequent reflective discussions with the second author (LL, who is independent of the participant population) throughout the data collection and analyses processes to ensure transparency and reflexivity [18].

### 2.8. Ethics Approval

Ethics approval for the study was obtained from the Darling Downs Health Human Research Ethics Committee for multi-sites (Reference HREC/2020/QTDD/69958). Subsequently, site-specific approvals were obtained from all the health services involved in the study.

## 3. Results

Twenty-three interviews were conducted with participants representing ten health professions (Table 1), yielding almost 15 h of audio-recorded data. Interview duration ranged between 28 and 60 min. Six interviewees were with clinicians that held only direct student supervisor roles, while the others were in roles that had student supervision, as well as leadership, management, or support functions related to student supervision. Participants’ experience in their respective professions ranged from eight to 48 years. Participants had been in their roles for between three months and 17 years, at the time of the study.

Four themes were constructed through the analysis process, namely compounding stress, negative impacts on student learning, opportunity to flex and innovate, and targeted transitioning support strategies.

### 3.1. Compounding Stress

Participants described supervising a student through the first wave of the pandemic as a time of great stress. At one level, they talked about being under a lot of stress as a community or family member trying to juggle home-schooling and facing day-to-day issues such as limitations in grocery supply in the supermarkets. Adding to this was the stress of being a frontline clinician experiencing changes in their duties related to service delivery. Being a student supervisor, and having the responsibility for a student’s welfare, on top of all the already abounding stress, meant that it was a very challenging time for student supervisors. Participants said:

*So, we all had our version of our own personal crisis and our own professional crisis but also making sure from a clinical education perspective that the student was feeling well supported…It was an exhausting time really and I feel like the whole team is still fatigued from it. We are all fatigued.* Int 13

*I think they (student supervisors) were already a bit escalated and fatigued and wearing masks, and personal life was disrupted too. They were thinking about ‘how can I meet my basic needs? Well, I have to worry about getting things from the supermarket. Everything is sold out. How do I feed my family because I can’t get groceries’…I think students were arriving on placements not wanting to come to placement because they were worried about coming to a health facility in the middle of the pandemic. There was also extra workload on the educators trying to work out new systems, and make sure they were working OK… So, I think for them, the added burden certainly was evident there*. Int 5

*This is the issue with the pandemic all around, we have practitioners who were feeling the same level of anxiety as the rest of the community…They’re anxious for all the same reasons that the community is anxious, and they’re having to deal with presentations around that.* Int 3

Participants attributed some of this stress and anxiety to the lack of clear communication. One participant noted:

*Communication generally now…has improved significantly in terms of advice being very clear…whereas back then it was kind of like ‘They said this, but they haven’t covered that. They haven’t explained what it means for the student who has been to X (nearby large city, a frequently dcelared hotspot)’. Some of the communication was lacking in clarity*. Int 7

This participant also went on to explain why communication around student placements was a complicated matter:

*The student placement problem is very complex, because there is the student, the clinical educator, the health service, the university, and the rural or regional community they are usually coming in and out of. There is a lot of players to then communicate in a very effective way. All these players are bound by their systems and their processes as well.* Int 7

Another participant noted the lack of inclusion about student placement considerations in the COVID-19 pandemic clinical response plan at the outset of the pandemic. They noted the importance of future pandemic plans including messaging around student placements and supervision, to enable seamless communication. They said:

*The COVID-19 pandemic plan had been sort of based around swine flu, and as far as I know, didn’t contain anything about student supervision, and was grossly inadequate to deal with a true global pandemic of a novel virus. I think we’ve all recognised that. So, yes, we will need to have a vast improved clinical response pandemic plan in the future to include management strategies and dealing with student supervision—that would be a very sensible thing to include*. Int 23.

### 3.2. Negative Impacts on Student Learning

Overall, participants agreed that the peak pandemic period in early 2020, and resulting impacts on health service delivery, had negative impacts on student learning during placements. This was in the form of reduced breadth and variety of clinical presentations, lack of face-to-face interactions with patients and team members, and lack of opportunities to consolidate learning. One participant with direct student supervision responsibilities recalled the feedback received from their student during the placement:

*Her (student) feedback at the time was that it was incredibly difficult, incredibly stressful, and she missed the opportunity of integrating the work with the clinical environment. In terms of her learning, that was a big loss for her…This was her opportunity to actually get hands-on practice, and then she was back working from home*. Int 3

Another interviewee, talked about the impact of PPE shortages on student learning:

*But students weren’t allowed in the operating theatre at the height of COVID, mainly for the purpose of preserving PPE. There was a lot of concern about a lack of supply of PPE and a lot of concern about what was going to happen when we got this big influx of patients we were expecting. The decision was made at the executive level to disallow students from coming into theatre. I think that was partly detrimental to their experience in orthopaedics…* Int 19

This participant also elaborated on the missed learning opportunities for students in an acute, orthopedic setting:

*…the patient contact and the patient interaction is very important. One of the key things we teach students in orthopaedics is how to examine patients’ joints, and if they don’t get that and they don’t practice that, that’s going to be detrimental to their career in general practice or emergency medicine or whatever*. Int 19

In addition to impacts on clinical skills and knowledge, it was also noted that students’ sense of professional identities were affected given the lack of face-to-face interactions and working in a team environment:

*I do have concerns about the students’ sense of professional identity because they were so isolated for so long, learning from home, they didn’t have opportunities with their cohort and their peers*. Int 5

Another participant, who held direct, as well as strategic student supervision roles, reflected on their experiences supervising students since the pandemic onset (early 2020) and further along around mid 2021. They noted significant continued impacts amongst students whose earlier placements were impacted in 2020. The participant said:

*We have had a large cohort of student come through where they have been required to meet entry level on the speech pathology assessment tool. So, thinking about what they lost last year, they were not exposed to a lot of face-to-face interactions with patients. Essentially, they had to learn how to be a speech pathologist on telehealth…This year, (i.e., 2021), we saw a lot of those foundational skills, the professional standards such as communication, professionalism, reasoning, that students would have been able to establish and cement, that was really missing because they lacked the face-to-face exposure in the previous year, 2020…It was a very steep learning curve for a lot of students we have just had*. Int 13

Echoing this observation, another participant holding a strategic leadership role said:

*Anecdotally, our clinical educators are saying that this year’s students, the 2021 students, don’t appear to be as well prepared. And that’s because we’ve just had a first semester of fourth year and final year students who probably had all their placement preparation online, plus in the restructuring of curriculums, this may be their first client-centred placement ever. So, some of them are coming to placement as final years not ever having interacted with a patient or a client. They’ve done project placement, role emerging placements in organisations such as the NDIS (National Disability Insurance Scheme).* Int 21

### 3.3. Opportunity to Flex and Innovate

Some supervisors, because of their changed workloads (e.g., cancellation of elective surgeries) were able to invest more time in teaching:

*But it also meant that I had a lot more time to teach students and to teach the residents. I spent a lot of time doing research. So, we had a very productive year last year as a department, research-wise*. Int 19

*I think this opportunity or disruption we have had has probably meant looking at things differently. I think that then did create that opportunity to say ‘we don’t have to do things how we have always done them’…Both the student and clinical educators have reflected that it was helpful all around, because we have got students that travel often, every weekend, from X (nearby large city)…Previously we have had models where if the students may be travelling back to X (nearby large city), we would say ‘You can leave at lunch time on Friday and work on your project over the weekend’. Whereas now, I think we would be more flexible about saying ‘Friday, don’t come into this health service. If you want to go home on Thursday night, you can. We don’t need you to be right here with this traditional model of us observing you all the time for you to be able to complete the goals of your placement’.* Int 7

*So we saw the structure, which I call the model of placement, be explored, and changed and then, within that, we saw changes to how students were being supervised within those models. Students, due to physical distancing requirements and the diminished client contact availability, we saw a lot more peer-assisted learning, peer-supported learning, supervision practice. A lot more students (were) being required to be active leaders and collaborators in their own learning and source-learning opportunities. We saw a shift to peer supervision…online supervision practices, telesupervision practices and we saw educators needing to quickly adapt the way that they structured their supervision sessions*. Int 21

One participant in a leadership role supporting students on placements talked about a new model implemented in their region where occupational therapy students were able to be employed in allied health assistant roles to assist with pandemic-induced service delivery pressures at a nursing home. Notably, the students were also engaged in some interprofessional practice in their new roles. The participant talked about the advantages of this model:

*The great thing about the students was we had already built rapport so that wasn’t something that we needed to do as with a new staff member. They were very confident to ring and ask for help and they knew the supervisor and myself, so we had that trust built in. They were out there representing the whole of allied health but obviously with an OT (occupational therapy) focus because that was their background. They created some allied health plans, and those plans were carried out while they were there. They ran groups, they participated in quality projects specifically around cognition and dementia.* Int 14

A further participant talked about adaptation of an interprofessional student-led health and wellness clinic from a face-to-face to a telehealth format:

*…That’s usually an interprofessional clinic where they run programs for people with rising risk of chronic disease…The participants will come in and see individuals for their professional input, but they’ll also do a supervised exercise program… They swapped that relatively quickly over to a telehealth-delivered model, particularly for the exercise program component… The students were involved in delivering the program and also in helping to adjust the program very quickly from face-to-face to telehealth instead.* Int 2

Another participant, who trialed a collaborative student research project placement with three vulnerable students to enable them placement opportunities as they could not undertake hands-on placements due to their own health challenges, talked about students’ feedback about this novel model:

*It was a cross-cultural group… We would talk through daily, via Microsoft Teams, for half an hour, generally more, what it was that they were learning, the connections that they were making, doing that stuff around linking theory with practice, honing those research skills, but also thinking about what the meant in terms of clinical practice. As a project, it worked really well. As a learning experience, they were very positive about that. They felt really well-supported. It gave them an opportunity to develop some nice teamwork skills*. Int 3

This placement model was able to be continued despite quarantine requirements, and one student continuing placement from overseas due to the inability to get back into Australia due to pandemic-induced travel restrictions.

### 3.4. Targeted Transitioning Support Strategies

Several participants talked about the need for healthcare organizations to devise targeted support strategies for new graduates whose placements were impacted in some way because of the pandemic. Provision of effective clinical supervision and creating more feedback mechanisms were some of the strategies proposed by participants. A participant that had strategic responsibility for workforce support, emphasized the need for supporting emerging new graduates in the professional skills area (i.e., beyond clinical skills and knowledge). They said:

*…but for the new graduates that are starting, it’s hard sometimes when you come in because you don’t know what you don’t know. Part of that may be getting feedback from their professional supervisor around not just straight clinical issues…but also about some of the fitting into the workplace and how things work here and other things that they may have been a bit light on because of changes to placements... For allied health, supervision is probably one of the primary mechanisms because it is something that everyone needs to have some sort of involvement in, and because it has a learning and development focus anyway, it’s easy to link those sorts of things in there*. Int 2

One partcipant highlighted the importance of clinical supervision in developing professional skills:

*I think we need to fundamentally go back to very strong supervision of the new graduates around professional skill development, clinical reasoning skill development, theory-to-practice development...* Int 21

Another participant stressed the importance of creating sufficient feedback mechanisms to identify what support needs new graduates have: “there are other things we could do which give them (new graduates) the opportunity to feedback more” (Int 16). Reinforcing this, one participant, in a strategic leadership role overseeing placements across multiple health services, said:

*We will probably need to keep a closer eye on them as they transition into practice; proactively be working with them to look at strategies, to make sure they are achieving that sense of professional identity and coping with the time and volume in the workplace, and the busyness…ow the pressures in the hospital system, the pressures on beds, the patient flow is really rapid and fast…high pace for them to be entering into and there’s lots of changes within the health setting, more than the usual. I think we do need to keep an eye on their wellbeing and coping as they transition into practice*. Int 5

One participant raised the importance of targeted support strategies to meet different individuals’ needs:

*I don’t think it’s going to be possible to have a blanket approach. Everyone is going to be different, and everyone’s situation is going to be different. At an organisational level, making sure that there’s good messaging around support services available, making sure that line managers are educated to check in on their staff.* Int 19

Another interviewee reinforced this and outlined practical strategies for healthcare organizations to consider:

*So, I think for employers sitting down with those new grads when they start and talking through their core experiences, and getting an idea of which students had competent experiences in different placements and where they are at, and then identifying where there are needs. So, for example, there could be an area where a (physiotherapy) student’s musculoskeletal placement, was all via telehealth. That’s really important for an employer to know. So, they can say, look, before we look at you starting that, we might have you do some work shadowing, spend some time with doing some of the competencies with our senior staff when you go into that clinical area, as opposed to having them start day one seeing six or seven patients. Because they haven’t had that face-to-face experience as a student.* Int 17

## 4. Discussion

This study explored healthcare workers’ experiences of supervising prequalification students on placements in rural healthcare settings at the initial onset of the pandemic. It has provided evidence on the additional challenges healthcare workers were exposed to in these resource-constrained settings, owing to the COVID-19 pandemic. It is unsurprising that student supervisors in healthcare settings experienced high levels of stress, anxiety, and worry, triggered by student supervision responsibilities. It is noteworthy that this cohort working in resource-constrained areas, already experience a number of barriers such as professional and social isolation, complex role duties, and cultural challenges [15,19]. Recruiting and retaining the healthcare workforce in these areas is crucial towards achieving equity in healthcare access to vulnerable populations, including the Aboriginal and Torres Strait Islander communities [3,20]. This becomes more crucial in these unprecedented times as these disadvantaged regions are said to experience higher COVID-19 mortality and morbidity rates [21]. Therefore, added mental health and wellbeing challenges pose a great risk to the retention of the healthcare workforce in rural areas, likely therefore, to further widen the inequalities in healthcare in these regions. It is more important now for healthcare organizations to devise, implement, and strengthen wellbeing measures for its workers servicing rural areas. Female healthcare workers may benefit from additional support measures given the higher risk of experiencing mental health and wellbeing challenges at work [22,23]. This study highlights the need for support measures targeting those who have the additional responsibility of student supervision.

This study has documented the impacts of the pandemic on student learning and support, and their career progression, from a supervisors’ point of view, across several professions. In line with previous findings, this study confirms that students on placement during the initial phase of the pandemic experienced several disruptions to their learning [2,3,4,24]. The findings from this study have addressed a gap by providing a triangulation point to validate student concerns with that of student supervisors [3]. Students and supervisors are both unified in their concerns about the student cohort whose placements have been impacted by the pandemic. Healthcare organizations and supervisors of new graduates need to be mindful of this as these students graduate and enter the workforce [3], especially in relation to professional skills (i.e., other than clinical skills and knowledge) such as communication, professional identity, and teamwork. As the findings of this study indicate, there is no one-size-fits-all approach to supporting new graduates. A combination of approaches is recommended including needs assessments, effective clinical supervision practices, mentoring, peer support systems such as buddying, social supports, and supportive and open work environments and cultures. Ardekani and colleagues [24] also proposed a schema of support system to better support students on placements during unpresented times. This includes a combination of academic and mental health supports [24].

Whilst a time of challenge, the COVID-19 pandemic has also provided an opportunity to adapt and innovate. Findings of this study showcase novel, innovative models of student placements, with some developed to cater to vulnerable students with health challenges who would have otherwise missed out on placement opportunities. Student supervisors in this study found numerous ways of being flexible and adaptable to meet student learning needs within the constraints of lack of PPE, physical distancing, and quarantine requirements. Some participants were also successful in facilitating interprofessional learning with students on placements. As previously suggested, the pandemic has provided a great opportunity for those involved in education of students, to examine, review, and strengthen curriculum and teaching practices [25]. New and innovative models and approaches trialed in student placements need to be evaluated and documented, so as to inform future directions. Future research could also investigate in more depth the clinical supervision parameters (i.e., structures and processes) impacted during the pandemic more specifically.

### Limitations

This interview study explored healthcare workers’ experiences of supervising students through the first wave of the pandemic. Only experiences of student supervisors and those in roles supporting them were studied. Student experiences and perspectives were not explored. While every effort was made to have good representation from all eligible health professions, recruiting participants from nursing and medicine was particularly challenging given the competing priorities and time pressures faced by these predominantly frontline clinicians. However, as sampling included representation from allied health, nursing, and medicine, the findings have been informed by healthcare workers from a broad range of professions. Since access to participants for the overall study was challenging owing to the pandemic pressures, participant checking of data was omitted in order to decrease participant burden during these challenging times. However, the researchers employed a variety of other measures to enhance the trustworthiness and rigor of the study.

## 5. Conclusions

This is the first study conducted across ten health professions to explore experiences of rural student supervisors’ supervision of students through the initial phase of the COVID-19 pandemic. Findings of this study confirm and validate the disruptions to learning and concerns of student transition to practice, already reported by studies investigating student experiences. Students impacted negatively during their placements because of the pandemic will benefit from additional targeted support strategies when they return to the health setting for subsequent placements, or to work as new graduates. Although an incredibly stressful time for supervisors, this period allowed them to trial innovative adaptations and approaches to teaching and learning practices, including new models of student placement. Technology was utilized to facilitate supervision of vulnerable students and those that were unable to physically return to the university. In some instances, students were utilized to fill a workforce gap to battle the pandemic. These novel and innovative practices and models need to be rigorously evaluated to inform future practice. Healthcare organizations need to invest in developing, implementing, and strengthening workforce support strategies so that staff can be retained in rural areas, and thus, not adversely affect healthcare quality and access in these regions.

## Figures and Tables

**Table 1 ijerph-19-05555-t001:** Participants’ professional background.

Profession	N (23)
Nursing	2
Midwifery	2
Medicine	3
Allied health	(16)
Exercise physiology	1
Nutrition and Dietetics	2
Occupational therapy	4
Physiotherapy	4
Radiography	1
Speech pathology	2
Social work	2

## Data Availability

Data and materials are protected by ethics but may be made available in de-identified format upon contact made with the corresponding author.

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
