# Peer review of "Impact of the COVID-19 Pandemic on Clinical Supervision of Healthcare Students in Rural Settings: A Qualitative Study"

_ijerph, 2022, doi:10.3390/ijerph19095555_

Round 1
Reviewer 1 Report
Thank you for your efforts. Though Interviewing health care professionals who were busy during the pandemic was not easy, you did some very difficult research.
The review results are as follows.
1. In qualitative research, researcher is a very important tool. In my personal opinion, it is judged that the subject of research results described by the reflective theory analysis method needs to be analyzed in more depth.
2. Even it is a practice course of students majoring in health care, the courses are diverse. Thus I hope that even in four institutions where collected data will have a brief explanation on how clinical practice is conducted at health care centers in rural areas
3. The title of this study “the considerations of student clinical practice in rural areas” is not revealed in the theme derived from the research results. I hope that the analysis can be done in depth so that the uniqueness of this paper can be revealed.
4. In the study results, the interview contents of the interviewees were presented and the interviewee numbers were presented for each content, but their general characteristics of the interviewees cannot be known in detail. I would like to be specific in the paper so that the characteristics of the 23 interviewees and their responses can be compared.
Author Response
Please see attached letter.
Reviewer 2 Report
Very good job, very well exposed and clear results.
I would improve the conclusions, they are too few for all the findings they propose.
Author Response
Please see attached letter.

Reviewer 3 Report
Dear Authors,
thank You very much for the opportunity to read such an interesting paper It shows challenges healthcare workers were exposed to the COVID-19 pandemic. This study has documented the impacts of the pandemic on student learning and support. Findings of this study are very important and confirm the disruptions to learning and concerns of student transition to practice.
I have some suggestions:
In introduction please add something about educating students of medical faculties (clinical education) a brief explanation to introduce for the reader unfamiliar with the education system in Austarlia
Please describe briefly how long the students were supervised,
Line 172 l is missing
Author Response
Please see attached letter.

Round 2
Reviewer 1 Report
Thanks for your efforts.
- I would like to present the simple demographic characteristics of interviewees (Int. 1~23).
- Your research produced 4 themes. Probably each theme can produce their sub-themes.
Author Response
Please see attached letter

Round 3
Reviewer 1 Report
Dear author,
This is the 3rd review, but the results of the previous comments are not reflected in the text. Please confirm the review opinion written in the 2nd comments.
Thanks.